

# Experimental exposure to urban and pink noise affects brain development and song learning in zebra finches (*Taenopygia guttata*)

Dominique A. Potvin[1,2,3], Michael T. Curcio[4], John P. Swaddle[4,5] and Scott A. MacDougall-Shackleton[2,3]

[1] Research School of Biology, Australian National University, Canberra, ACT, Australia
[2] Advanced Facility for Avian Research, University of Western Ontario, London, ON, Canada
[3] Department of Psychology, University of Western Ontario, London, ON, Canada
[4] Institute for Integrative Bird Behavior Studies, College of William and Mary, Williamsburg, VA, United States
[5] Centre for Ecology and Conservation, University of Exeter, Exeter, United Kingdom

## ABSTRACT

Recently, numerous studies have observed changes in bird vocalizations—especially song—in urban habitats. These changes are often interpreted as adaptive, since they increase the active space of the signal in its environment. However, the proximate mechanisms driving cross-generational changes in song are still unknown. We performed a captive experiment to identify whether noise experienced during development affects song learning and the development of song-control brain regions. Zebra finches (*Taeniopygia guttata*) were bred while exposed, or not exposed, to recorded traffic urban noise (Study 1) or pink noise (Study 2). We recorded the songs of male offspring and compared these to fathers' songs. We also measured baseline corticosterone and measured the size of song-control brain regions when the males reached adulthood (Study 1 only). While male zebra finches tended to copy syllables accurately from tutors regardless of noise environment, syntax (the ordering of syllables within songs) was incorrectly copied affected by juveniles exposed to noise. Noise did not affect baseline corticosterone, but did affect the size of brain regions associated with song learning: these regions were smaller in males that had been had been exposed to recorded traffic urban noise in early development. These findings provide a possible mechanism by which noise affects behaviour, leading to potential population differences between wild animals occupying noisier urban environments compared with those in quieter habitats.

## INTRODUCTION

Research into the effects of urban noise on the behaviour of wild animals has increased over the past decade. We now know that several species of birds change their song in association with anthropogenic noise. Specifically, birds appear to alter the frequency (*Brumm, 2006a.*; *Potvin, Parris & Mulder, 2011*; *Slabbekoorn & Peet, 2003*), amplitude (*Kight & Swaddle, 2015*), timing (*Brumm, 2006b*; *Cartwright et al., 2014*; *Fuller, Warren & Gaston, 2007*),

Corresponding author
Dominique A. Potvin,
dominique.potvin@anu.edu.au

meme use (*Cardoso & Atwell, 2011*; *Potvin & Parris, 2013*), and tempo (*Potvin, Parris & Mulder, 2011*; *Slabbekoorn & Den Boer-Visser, 2006*) of their songs in noisy environments.

While some of these adjustments are immediate responses to the sound environment (*Halfwerk & Slabbekoorn, 2009*; *Kight & Swaddle, 2015*; *McMullen, Schmidt & Kunc, 2014*; *Potvin & Mulder, 2013*; *Verzijden et al., 2010*), others are more consistent with longer-term improvements of signal efficacy in noisy environments—in agreement with the Acoustic Adaptation Hypothesis (*Morton, 1975*). For example, the differential occurrence in noisy areas of particular memes or dialects is evidence of cross-generational cultural evolution (*Luther & Baptista, 2010*; *Luther & Derryberry, 2012*; *Potvin & Parris, 2013*). However, the proximate mechanisms that contribute to this process—that is, the developmental, physiological, neurological, or mechanical changes resulting in song differences between generations—are still unknown.

One theory suggests that noise may be a source of developmental or chronic stress (*Wright et al., 2007*). Birds living in noisy areas may chronically engage their stress response, allowing them to cope with the stressor but pay a longer-term cost. While there is currently some evidence for this, results have been inconsistent (*Blickley et al., 2012*; *Bonier, 2012*; *Crino et al., 2013*; *Partecke, Schwabl & Gwinner, 2006*). If chronic stress is affecting young songbirds in noisy environments, cognitive development may be affected, resulting in altered songs (*Buchanan et al., 2004*; *Nowicki, Searcy & Peters, 2002*; *Schmidt et al., 2013*; *Spencer & MacDougall-Shackleton, 2011*). This would likely have an impact on the syllable content of songs (*Brumm, Zollinger & Slater, 2009*; *Schmidt et al., 2014*; *Schmidt et al., 2013*; *Spencer et al., 2003*; *Zann & Cash, 2008*). Testing this idea would require documenting connections among anthropogenic noise, biomarkers of developmental chronic stress, and adult song.

Even if birds do not respond to noise as a chronic stressor, noise may still affect the song-learning process by affecting neural and cognitive development and function (*Iyengar & Bottjer, 2002*; *Kujala & Brattico, 2009*). For example, changes in neuroanatomy and song learning may occur through the interruption or masking of tutor-tutee communication, through the impairment of auditory feedback during the song-learning process, or through other as yet unknown mechanisms (*Dooling & Blumenrath, 2013*; *Kight & Swaddle, 2011*). In the well-studied model of zebra finch (*Taenopygia guttata*) song learning, tutees (young males) form an auditory memory of songs sung by tutors (often their fathers) during a period of approximately 20 days (post-hatch day (PHD) 15–35), then subsequently attempt to match their vocal output to these auditory memories through the production of subsong (PHD 35–50) and plastic song (PHD 50–80; *Catchpole & Slater, 2008*). Noise could affect this process in multiple ways. First, noise may mask components of tutor song that may then not be learned by tutees, or affect the father's singing rate. Since anthropogenic noise is known to interfere with parent–offspring communication in other contexts (*Leonard & Horn, 2008*; *McIntyre, Leonard & Horn, 2014*; *Schroeder et al., 2012*) it is reasonable to assume that song learning may also be affected by such interference. Second, noise may disrupt parental feeding rate or incubation (*Potvin & MacDougall-Shackleton, 2015b*) and thus indirectly affect development. Third, noise may alter social interactions among birds, and thus affect song learning. For example, urban noise can alter social spacing

in parid songbirds (*Owens, Stec & O'hatnick, 2012*). Finally, noise may impair auditory feedback during the production of subsong and plastic song by tutees and thus affect song development (*Tschida & Mooney, 2012*; *Zevin, Seidenberg & Bottjer, 2004*). Individual song variation is higher in some wild birds occupying noisy environments, consistent with impaired auditory feedback during song learning (*Gough, Mennill & Nol, 2014*).

We aimed to experimentally determine if noise affects song learning, song development, and development of the song-control regions of the brain. We used both field recordings of urban noise (traffic and other urban sounds) as well pink noise (white noise filtered to 0.1–3 kHz) in order to determine if noise effects are specific to particular sounds or general to noise in a particular frequency range. The replication of the general experimental design across two labs, although using different birds and different noise profiles in each experiment, also let us investigate more robustly whether there are generalities in the ways zebra finches respond to ambient noise. We also aimed to determine whether noise may act as a chronic developmental stressor and have long-lasting effects on circulating glucocorticoid hormone levels. By conducting an experiment under laboratory conditions we isolated the effects of noise (both recorded urban noise and pink noise) and controlled for other characteristics associated with urban habitats that may induce a stress response or affect song-learning in the wild, such as lighting (*Kempenaers et al., 2010*), breeding density (*Hamao, Watanabe & Mori, 2011*), diet (*Gavett & Wakeley, 1986*), or parasite load (*Bonier et al., 2007*). If nestling birds perceive anthropogenic noise as a chronic stressor, we predicted that noise would have long-lasting effects on baseline corticosterone levels. This in turn could affect development of song-control brain regions, resulting in under-developed song-control regions in noise-treated birds. Furthermore, we predicted that noise would hamper the development of song among juvenile male zebra finches either due to developmental stress, interference with auditory feedback, and/or masking tutor songs. If noise creates developmental stress we predicted that, similar to previous studies, songs of birds reared in noise would be developmentally delayed, would have reduced similarity to tutors' songs, and would have fewer distinct syllable types (i.e., lower complexity). If song masking and/or auditory feedback disruption occurred, we predicted that tutees subjected to noise would sing higher frequency songs (reducing the effect of masking on lower frequencies) compared to tutors and tutees learning under quiet conditions.

## METHODS

We conducted two independent experiments testing the effects of noise on song learning in zebra finches. In study 1, we measured the effects of experimental exposure to recorded urban noise on song learning, corticosterone levels, and the size of song-control brain regions. In study 2, we measured the effects of experimental exposure to synthetic noise in the frequency range of urban noise (white-noise 0.1–3 kHz band-pass filtered) on song learning. The timelines of these two studies are illustrated in Fig. 1.

**Group 1: Traffic noise**

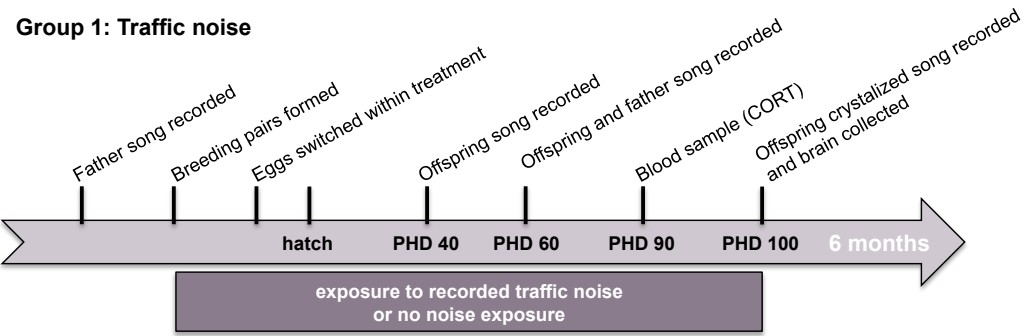

**Group 2: Pink noise**

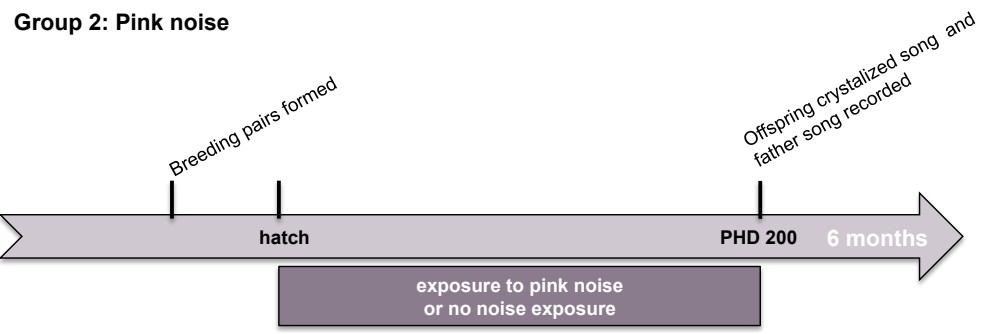

**Figure 1** **Timeline of experiments.** Stylized timelines of the two noise-exposure studies. The timelines are not to scale.

## Experimental protocol

### Study 1: urban noise

We sourced twenty breeding pairs of zebra finches from a previously established domestic flock at the Advanced Facility for Avian Research at the University of Western Ontario These birds were also part of parallel studies on the effects of noise on adult song and nestling growth rates and survival (*Potvin & MacDougall-Shackleton, 2015a*; *Potvin & MacDougall-Shackleton, 2015b*).

We housed one male and one female in a cage in acoustic isolation chambers (modified audiometric testing booths, see *Potvin & MacDougall-Shackleton, 2015b*), along with 2–3 other breeding pairs (in separate cages that were visually isolated from one another and on different shelves arranged vertically) for six months. Each chamber was 70 cm wide, 90 cm deep and 172 cm high (interior dimensions) with vertical shelves. Lighting was mounted along one of the walls to provide consistent illumination to all shelves. We thus had 20 breeding pairs held in six chambers. We gave all birds water and premium finch seed *ad libitum*, along with a daily tablespoon of eggfood (boiled egg, bread and cornmeal mixture). Birds were kept on a 14:10 h light:dark photoperiod at approximately 22 °C to maintain birds in a breeding state. Three of the chambers (10 zebra finch pairs) were treated as a control group ("silent" group; no noise played), while the other three chambers (10 pairs) were exposed to urban noise during all daylight hours. Urban noise was recorded by DAP at a busy urban park (Melbourne, Australia), using the recording equipment described

below, and was played randomly interspersed with unedited soundtracks of trains, cars, motorcycles, and lawnmowers downloaded from Soundbible.com. We thus used a total of 9 tracks of urban noise, each between 1 and 10 min long, that were played in random order throughout daylight hours. This urban soundtrack was played inside the chambers using an iPod touch connected to amplified computer speakers (Logitech S11). A power spectrum of the noise used (over a period of one hour) is provided in *Potvin & MacDougall-Shackleton (2015a)*. Noise levels varied over time during the track, replicating variation experienced in urban parks. Sound pressure levels were regularly checked using a Realistic 332050 Sound Level Meter, using A weighting. Each testing booth contained 3–4 cages (pairs), one of which was adjacent to the speakers (lower shelf), with noise levels ranging from 60–80 dBA SPL at the centre of the cage; another one (or two, in the case of chambers that contained four cages) which was placed at a mid-range distance (middle shelves) from the speakers (average background noise range 50–75 dBA SPL at cage centre), and the last cage furthest from the speakers (top shelf) , which experienced noise at an average sound level of 40–70 dBA SPL. We therefore generated four treatment groups for subsequent statistical analyses: Loud Noise, Moderate Noise, Soft Noise, and Silent (no urban noise). Because the cages were small relative to the dimensions of the chamber the variation in noise amplitude within a cage was negligible.

Of the twenty pairs, four did not reproduce successfully, and one further pair only produced female young. All other nests ($N = 15$) successfully fledged at least one male nestling, resulting in 24 juvenile males total that could be included in the study.

*Cross fostering.* To reduce genetic and maternal effects on our dependent measures we used a randomized cross-fostering protocol with nests that were synchronous for the date of first egg-lay. This involved moving one egg from each (donor) nest into a paired (recipient) nest in another chamber, and vice versa (reciprocal cross-fostering so as not to alter the brood size of either nest). All nests were subject to cross-fostering. For nests with more than four eggs, a second egg was to be cross-fostered, however due to chance (only two nests with larger brood sizes were in fact synchronous) we were able to do a second cross-fostering only once. Cross-fostering was done both within and among treatments. As we could not track which individual was cross-fostered in each nest, we did not attempt to account for genetic relatedness in analyses.

*Song recording.* We recorded the tutors' (fathers') songs when their offspring were PHD 60 (Fig. 1, *Potvin & MacDougall-Shackleton, 2015a*). Each tutor male was isolated in a sound attenuation chamber for a period of 24 h, after which an adult female was introduced, inducing males to sing in all cases. These songs were recorded for five minutes using a Marantz Solid State PMD671 recorder and a Sennheiser ME67 directional microphone

The songs of all juvenile males (tutees) were recorded at PHD 40, PHD 60, and PHD 100 (Fig. 1). These time periods were selected as representative of the three major song-learning stages in zebra finches, with the aim of recording an example of sub-song (PHD 40), plastic song (PHD 60), and crystallized song (PHD 100) (*Catchpole & Slater, 2008*; *Slater, Eales & Clayton, 1988*). For the sub-song and plastic song recordings we placed the focal male in a

smaller cage (∼30 × 25 × 25 cm) adjacent to the original larger cage (∼60 × 40 × 40 cm) that contained his parents and siblings, all inside an acoustic chamber and turned off the urban noise. In this manner we could use a directional microphone (Sennheiser ME67) to record the focal male's vocalizations without measuring the vocalizations of the other birds. We recorded males using this protocol for 2–4 h during the hours of 09:00–13:00, and checked that all recordings had examples of subsong or plastic song, before putting males back into their original home cages and chambers, and turning the noise back on. For PHD 100 song, we used the same protocol to record the crystallized song from the juvenile males as used for the adult males prior to the experiment (see above).

*Corticosterone assay.* On PHD 90, we took a small blood sample from each male offspring ($N = 24$) by puncturing the brachial vein using a 26-gauge needle and collecting approximately 50 µL of blood into a heparinized microhematocrit capillary tube. We predicted that if noise acted as a chronic stressor to offspring then baseline corticosterone levels would be elevated. All blood samples were collected within three minutes of opening the door of the isolation chamber, therefore ensuring that we could analyze baseline corticosterone levels rather than acute stress responses associated with disturbance and handling (*Romero & Reed, 2005*). Blood was then centrifuged at 13G for 10 min and the supernatant plasma was collected and then kept frozen (−30 °C) until assay. Plasma was assayed for total corticosterone using a specific and sensitive radioimmunoassay kit (ImmuChem 07-120103, MP Biomedicals, Orangeburg, NY, USA). All samples were measured in a single assay. Sensitivity of the assay was 12.5 ng mL$^{-1}$ and within-assay coefficients of variation were acceptably low at 9.6% and 3.9% for low and high controls.

### Study 2: pink noise

At a separate location (College of William and Mary, Williamsburg, Virginia, USA), a second group of zebra finches was subject to similar protocols (Fig. 1). Twenty-four pairs of zebra finches were housed in breeding cages (34 × 39 × 75 cm), randomly selected from a large outbred stock population, in two separate rooms (12 pairs in each room). Both rooms experienced a 14:10 light:dark photoperiod at approximately 20 °C, and were identically set up (room effects on reproductive success and other physiological factors have been previously tested and ruled out; J Swaddle, 2014, unpublished data). All cages were visually but not acoustically separated from each other within each room. Each pair was provided with Volkman Avian Science Super finch seed, grit, cuttlebone, and vitamin-enriched (Vitasol) drinking water ad libitum, as well as two wooden perches, a plastic nest box and sufficient hay for nest building. Breeding checks were conducted every other day, and the number of eggs and hatchlings was recorded throughout the experiment.

In the experimental room, a small speaker (Memorex ML622) was attached to the back of each cage in the center and connected to an mp3 player (Sandisk Sansa). Noise was played through each speaker starting on PHD1and continued for the remaining duration of the experiment. The treatment noise was a 0.1–3 kHz pink noise (white noise bandpass filtered at 3 kHz), played back at 75 dBA SPL at the center of each cage for 24 h per day. Speaker functioning was checked every other day and amplitude of the noise was confirmed

every two weeks with an Extech instruments Digital Sound Level Meter (407727), using A weighting. The control room had some background noise from the surrounding animal facility, but this remained between 50 to 55 dBA SPL (measured in the center of each cage) throughout the study.

Birds were housed in these conditions for six months and allowed to breed throughout. All offspring produced were banded with numbered metal bands before fledging. Female offspring and female parents were removed after the first clutch in that cage had fledged. All pairs except for three (two in the noise treatment, one in the control room) produced a viable clutch. In total the pairs in the experimental (noise) room produced 29 male offspring across 8 pairs (i.e., four pairs did not produce male offspring). The pairs in the control room produced 28 male offspring across 7 families (five pairs did not produce male offspring). From these male offspring, we were able to record songs at PHD200 from five in each of the noise and control treatments, where each male came from different parents. The sample size was lowered because of premature deaths and occasional failure to solicit sufficient song on PHD200. On the day following each offspring (tutee) recording we also recorded their fathers (tutors). Song recordings followed similar protocols as described above. A male was placed in a quiet room (ambient noise < 50 dBA SPL) in a small cage (approximately $20 \times 20 \times 30$ cm) adjacent to an unrelated adult female in a separate small cage. Using a directional microphone (Sennheiser ME67) we recorded 10 clear directed songs from each male (tutees and tutors) onto a Marantz PMD661MKII recorder.

### Song analysis (both studies)

We used RavenPro 1.4 software (Cornell Lab of Ornithology) to create spectrograms of all recordings in order to identify and extract 5 random examples of song from each tutor ($N = 25$) and tutee ($N = 24$ at PHD 100 and $N = 10$ at PHD 200) song recording. We also used RavenPro to identify all periods of singing behaviour in the PHD 40 and PHD 60 recordings for Study 1 birds.

For juvenile subsong (PHD 40) we visually identified the number of fully-formed distinct syllables by comparing all syllables in the subsong to those in the same individual's crystallized song. We used the number of these crystallized syllables that were present in PHD 40 subsong as an indicator of song development (*Tchernichovski & Mitra, 2002*; *Tchernichovski et al., 2001*). We also used RavenPro to measure the minimum (lowest) frequency and maximum (highest) frequency of subsong over the entire PHD 40 recording, using the minimum and maximum frequency peaks at a threshold of >30 dB as identified by power spectra (*Beecher, 1988*).

For plastic song (PHD 60) we used Sound Analysis Pro 2011 software (*Tchernichovski et al., 2000*) to compare each juvenile male's PHD 60 song to their crystallized song (PHD 100). For crystallized song (PHD 100 and 200) we used the same software to compare each male's song to its respective tutor's (social father's) song. We ran a similarity batch analysis using an $M \times N$ matrix to compare all possible combinations of song-pair comparisons (5 from tutor compared with 5 from tutee), giving an output of estimates of song-similarity. We used the following estimates: % similarity (the percentage of tutors' sounds included in the final tutee song), accuracy (the similarity of each sound produced within songs

between tutor and tutee), % sequence similarity (the similarity of the tutor and tutee sequence of sounds within the song), and pitch difference. We used the mean estimates of similarity for each individual tutee in subsequent statistical analyses. The social father was assumed to be the tutor for all young, because zebra finches prefer to learn songs from birds with which they can socially interact, and the father was the only adult male visually present throughout the song-learning phase (*Clayton, 1987*; *Eales, 1989*; *Williams, Kilander & Sotanski, 1993*). However, in order to confirm this assumption, we compared similarity scores between sons and fathers with those between juvenile males and adult within the same chambers but from different cages, and adults from other chambers. Results showed there was a much higher similarity score between sons and fathers' songs than either of the other male groups, indeed verifying the vaildity of this assumption (Average similarity score between father and song = 54.58%; within chamber = 46.46%; between chambers = 45.84%).

For the offspring male crystallized songs (i.e., PHD 100 and 200), we also extracted the following song parameters independently using RavenPro. The number of notes per song and song complexity (number of different note types) were counted manually. Minimum frequency, maximum frequency, peak frequency (the frequency with the most energy), and song duration were measured in RavenPro using power spectra and spectrograms, using a power threshold of >20 dB. The cursors in RavenPro were placed at frequency and power thresholds to measure these values. Tempo (notes per second) was calculated using the number of notes and song duration. These crystallized song analyses were conducted for Study 1 and Study 2 groups separately.

All song measurements were made by one author (DAP) who was blind to treatment during analyses.

## Brain histology and analysis

Once juvenile males in study 1 were recorded on PHD 100, they were euthanized by an overdose of isoflurane and their brains extracted immediately from the skull. Brains were fixed by storing them in 4% paraformaldehyde for 24 h, then cryoprotected in 30% sucrose (in phosphate-buffered saline, PBS) for 48 h. They were then frozen on powdered dry ice and kept at −80 °C until sectioning. We sectioned one hemisphere (left or right randomly selected; sagittal plane, 30 μm sections) using a cryostat, collecting every second section into 0.1M PBS, then mounted sections onto microscope slides. We Nissl-stained the sections with thionin, then serially dehydrated them in graded ethanol solutions, cleared the sections in solvent (Neo-clear) then affixed a coverslip onto the slide with Permount (Fisher Scientific). Slides were subsequently examined under a Zeiss Axiophot microscope and photomicrographs of the song-learning brain regions Area X , HVC, and RA (robust nucleus of the arcopallium) were captured with a Spot Insight 5-megapixel microscope camera. These song-control regions were selected for analysis because they have previously been linked to individual differences in song and have been studied in the context of developmental stress. Images of the entire telencephalon were captured using a high resolution (2400 dpi) flat-bed scanner with transparency adapter. To calculate the volume of the song-control regions as well as the telencephalon as a whole we traced the

cross-sectional area of the regions of interest using ImageJ software (*Schneider, Rasband & Eliceiri, 2012*) and volumes were calculated by combining the cross-sectional areas and the sampling interval (60 μm) using the formula for a frustum (truncated cone). Any sections that were damaged or missing were accounted for by increasing the sampling interval appropriately. All tracing was done blind to treatment group.

## Statistical analyses

We performed all statistical analyses using a Bayesian framework in WinBUGS 1.4.3. To determine if noise affected offspring baseline corticosterone (study 1) we created a regression model to estimate the effect of noise treatment (silent, quiet, moderate, and loud noise) on baseline corticosterone levels, including uninformative priors (*McCarthy, 2007*). Since the number of siblings in a nest affects nestling condition, and therefore may also affect brain and song development in zebra finches (*Gil et al., 2006*) we also included number of brothers as a covariate. We estimated the mean and standard deviation from 200,000 samples from the posterior distribution, discarding the first 100,000 samples as a burn-in, and used the 95% credible intervals (CI) for our estimations. Following common Bayesian statistical procedures, we considered effects important if their 95% CIs did not overlap zero or if the 95% CIs were highly skewed and effect sizes were large (*McCarthy, 2007*).

We used a similar model to estimate the effect of noise on all song variables of interest (Study 1: number of crystallized syllables at PHD 40, similarity measurements of PHD 60–100 and PHD100 song minimum, maximum and peak frequency, bandwidth, duration, number of notes, complexity and tempo; Study 2: PHD200 song minimum, maximum and peak frequency, bandwidth, duration, number of notes, complexity and tempo). We also used a similar model to test for effects of noise on total telencephalon volume and relative volumes of RA, HVC, and Area X (volume of structure divided by telencephalon volume). To confirm our results, we repeated analyses on brain structures using the absolute brain structure volume with total telencephalon volume minus the structure volume as a covariate.

To determine whether song similarity to father was predicted by noise exposure, by brain structure volumes, or any interaction effects, we ran similar regression models for the birds in study 1 using the following independent variables: noise treatment group, telencephalon volume, RA relative volume, Area X relative volume, HVC relative volume and number of brothers. We used the DIC (Deviance Information Criterion) tool in WinBUGS to compare all models and determine the model that best predicted the variability in song similarity between tutor and tutee (lowest DIC by at least 2; *Spiegelhalter et al., 2002*).

## ETHICAL NOTE

All birds in study 1 were kept and treated in accordance with guidelines set by the Canadian Council on Animal Care (*Neil & McKay, 2003*), and all procedures in this study were approved by the University of Western Ontario Animal Use Subcommittee (protocol number 2007-089). Study 2 protocols were approved by the College of William and Mary Institutional Animal Care and Use Committee (IACUC-2012-11-23-8173-jpswad).

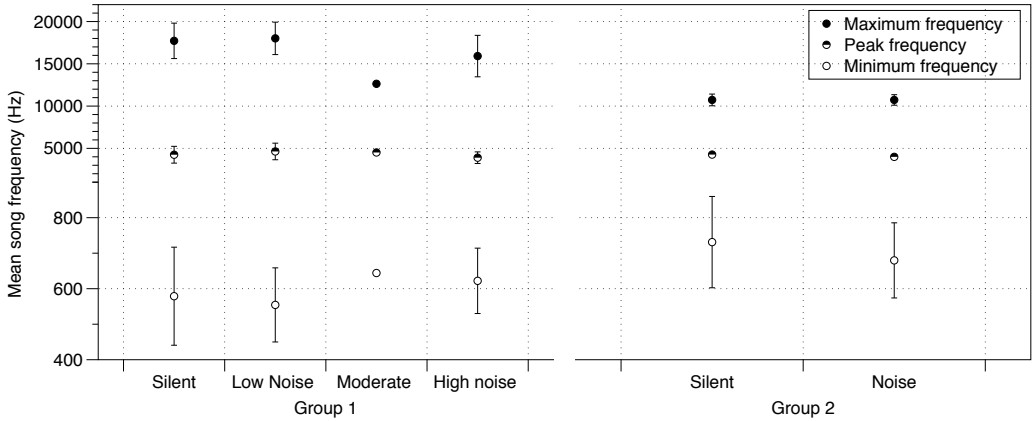

**Figure 2 Noise effect on song frequency.** Effect of noise treatment during development on crystallized song frequency characteristics (maximum, peak and minimum frequencies) in each of the two groups of zebra finches in the study. Error bars denote 95% confidence intervals.

## RESULTS

On PHD 40, treatment noise did not affect the number of fully-formed syllables (Table S1). However, both minimum frequency (mean effect = 37.51 Hz; 95% CIs = −14.4, 84.72) and to a greater extent maximum frequency (mean effect = −258.0 Hz; 95% CIs = −320.3, −196.1) were affected by treatment at this early subsong stage, as indicated by the large skew in CI. The effect of noise treatment on minimum frequency was primarily driven by the group experiencing the loudest noise, as they sang approximately 113 Hz (approximately 30%) higher than the other groups. Maximum frequencies were approximately 774 Hz lower (approximately 6%) in subsong sung by males from the moderate and loudest cages compared to those in the silent or quiet cages.

Songs recorded at PHD 60 had high similarity to songs recorded at PHD 100, demonstrating well-developed song by PHD 60. Similarity measurements at these stages were unaffected by treatment, indicating songs were developing at the same rate in birds across all treatment groups (Table S2).

There was no effect of noise on a variety of parameters of crystallized songs recorded at PHD 100 including the number of notes in a song, song duration, or tempo (Table S3). Additionally, the effect of noise on lowest frequencies identified at PHD 40 was no longer detected at PHD 100 (Table S3, Fig. 2). Maximum frequency was slightly lower in songs sung by males from the moderate and loud noise cages than those in the quiet or silent cages (mean = −874.9 Hz; 95% CIs = −934.8, −814.7; Fig. 2). Peak frequency showed a similar trend, with the loudest cages having the lowest peak frequency, although again the effect was small (mean = −69.81 Hz; 95% CIs = −129.5, −9.972; Fig. 2).

The PHD 200 songs of birds in study 2 were also affected by noise. Similar to study 1 birds at PHD100, there were no differences in minimum frequency (Table S4, Fig. 2). However, while peak frequencies were also lower in this noise group (mean = −228.9 Hz; 95% CIs = −404.3, −54.08; Fig. 2), birds in the noise treatment in study 2 sang higher,
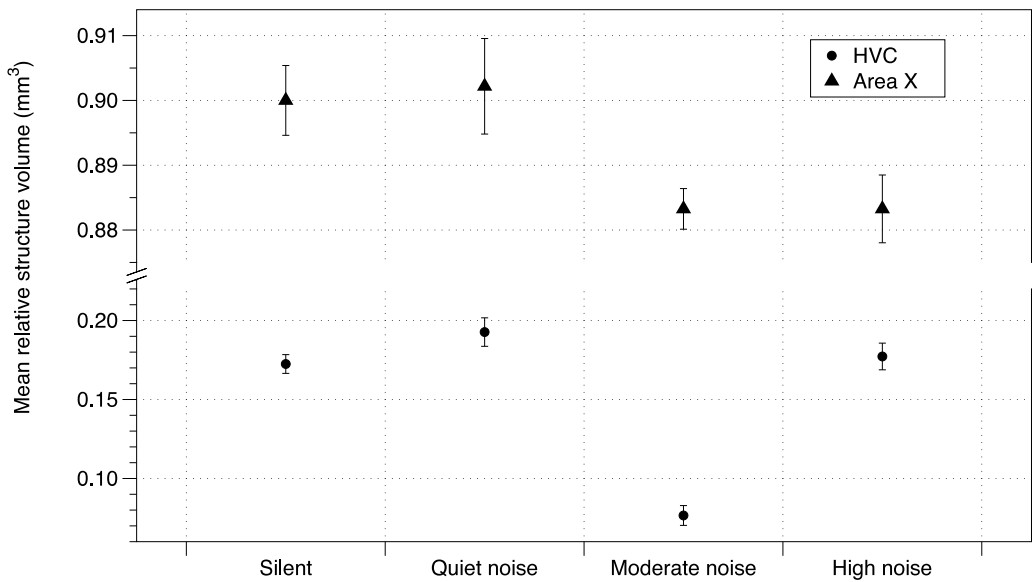

**Figure 3** **Noise effect on HVC and Area X regions of brain.** Mean brain structure volume (HVC and Area X) relative to total brain size from zebra finches in study 1 under each urban noise treatment condition. Error bars denote 95% confidence intervals.

not lower, maximum frequencies (mean = 117.7 Hz, 95% CIs = −17.65, 271.1; Fig. 2; all effects in Table S1).

Baseline corticosterone of offspring was not affected by the noise treatment (Table S5). While telencephalon volume and RA were unaffected by treatment, the noise treatment negatively affected HVC volume and Area X volume (using either method of correction for total telencephalon volume). The number of brothers—balanced between treatment groups (*Potvin & MacDougall-Shackleton, 2015b*)—also had a negative impact on brain structure volume (HVC mean = −0.0338 mm$^3$; 95% CIs = −0.057, −0.010; Area X mean = −0.011 mm$^3$, 95% CIs = −0.237, 0.017; Fig. 3; all effects in Table S6).

DIC analysis identified the model incorporating noise treatment, number of brothers, and Area X volume as being the model with best fit for all three measures of song similarity to father (% Similarity DIC score = 217.225; Accuracy DIC score = 196.356; % Sequence similarity DIC score = 219.638; all other scores for comparison in Table S7A). Area X itself was not important in the models predicting overall % Similarity or Accuracy, however it was important in predicting % Sequence similarity as was noise (larger Area X and higher noise levels were both correlated with lower % Sequence similarity), although their interaction was not important (Table S7B, Fig. 4). DIC penalizes models including redundant predictors, however this model produced the lowest DIC score even with cross-correlated values (Area X and noise, as above) (*Spiegelhalter et al., 2002*).

At PHD 200 for study 2 birds, although we did not have brain measurements for this group of birds, we identified noise as having an effect on % Sequence similarity between tutor and tutee (mean = 12.05%, 95% CIs = −1.055, 25.07) with other similarity measurements being unaffected (Fig. 4, all effects in Table S8).

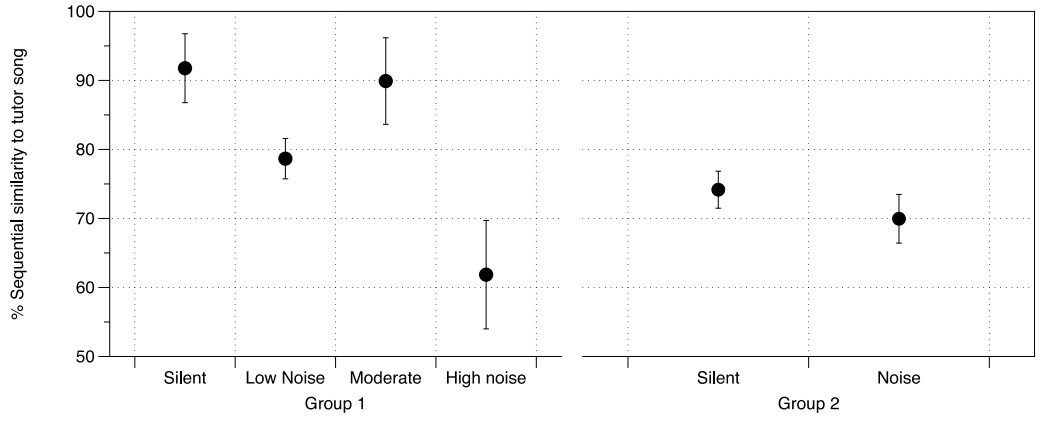

**Figure 4** **Noise effect on sequential similarity between tutee and tutor song.** Mean percent sequential similarity of tutee's crystallized song to tutor's song in each of the two groups of zebra finches in the study. Higher values denote a better copy of the sequence of syllables (syntax) by the tutee. Error bars denote 95% confidence intervals.

## DISCUSSION

We found that while noise during development did not affect baseline corticosterone in young male zebra finches, it did affect HVC and Area X volume—brain regions that are crucial to song learning. These neuroanatomical effects were accompanied by behavioural consequences. The similarity of song between the tutor (father) and tutee (son) was decreased by the combined effects of noise treatment, Area X volume, and number of brothers. Specifically, the similarity in the sequence of notes in a song, comparing father to son, decreased with increasing noise. This latter result was observed in two independent experiments (i.e., in both studies 1 and 2) with different populations of zebra finches, suggesting that the effects on song learning result from noise below 3 kHz rather than traffic and other urban sounds *per se*. In contrast to predictions made from observations of free-living birds singing in urban habitats, songs that developed in the noise treatments were consistently lower in peak frequency, and not higher in frequency range or minimum frequency. Furthermore, maximum frequencies showed inconsistent changes in response to noise treatments—in study 1 crystallized songs (at PHD100) had a lower maximum frequency, whereas in study 2, songs (at PHD200) had a higher maximum frequency. These mixed results make it difficult for us to support the hypothesis that putatively adaptive frequency changes observed in wild urban birds are due to an effect of noise on song development in early life stages.

While we attempted to identify whether chronic stress might be a mechanism by which noise affects song development in birds living in artificially noisy environments, our results do not show that baseline corticosterone in young birds was elevated under such conditions. Similarly, a previous study on nestling white-crowned sparrows (*Zonotrichia leucophrys oriantha)* found that young birds experiencing chronic traffic noise had lower baseline glucocorticoid levels than those in quiet conditions (*Crino et al., 2013*), while another showed similar results to our own (*Heiss, Clark & McGowan, 2009*). At the moment, most studies on the effects of anthropogenic noise on corticosterone levels have been conducted

on wild adult birds, and have produced varied results. While there is some evidence that certain species might experience chronically elevated glucocorticoid levels in urban or noisy areas (*Blickley et al., 2012*; *Bonier et al., 2007*; *Zhang et al., 2011*) other species do not (*Fokidis, Orchinik & Deviche, 2009*; *Partecke, Schwabl & Gwinner, 2006*; *Potvin & MacDougall-Shackleton, 2015a*). Many of these studies have attempted to isolate the effect of noise from other anthropogenic impacts on birds that might induce chronically elevated glucocorticoids; however it is clear that further research is needed to better understand how some species might be better able to acclimate or adjust their stress response to chronically noisy environments than others, and whether or how this acclimation may depend on age and social or genetic environment.

We did find an effect of noise environment on male brain structures associated with song learning. Telencephalon volume was unaffected, as was RA volume, but both Area X and HVC volumes were proportionally smaller in males from the noise treatment. The size of song-control brain regions is often correlated with song quality within- and between-species. We found that Area X, in particular, was related to the similarity of experimental males' song (at PHD 100) to their fathers' songs, along with noise treatment and number of brothers. Finding an effect of noise on corticosterone levels might have provided a mechanism by which noise could impact the size of Area X and HVC (*Buchanan et al., 2004*; *Schmidt et al., 2013*). However, it is likely that HVC is sensitive to environmental factors that may not instigate a chronic elevation in corticosterone. For example, noise may have been only transiently stressful to the birds at times other than those at which we sampled. Alternatively, reduced singing behaviour itself (though not measured in the present study) may have led to altered brain development. Noise and deafening has been shown to affect auditory and song learning circuits in previous studies of zebra finches (*Iyengar & Bottjer, 2002*), and neural plasticity of HVC is regulated by singing and social housing in canaries (*Alward et al., 2014*). We find it unlikely that noise exposure directly affects development of the song-control brain regions. Indeed, it is possible that noise may have affected incubation and/or parental feeding rates in study 1, as hatching rates were lower and nestling mass was lower in nests exposed to urban noise (*Potvin & MacDougall-Shackleton, 2015b*). Interestingly, no similar trends of breeding depression were found in Study 2 (D Potvin & S MacDougall-Shackleton, 2015, unpublished data). Determining the mechanisms by which noise affected neural and song development in our study would require further experiments, possibly with the manipulation or monitoring of food intake and/or real-time song learning behaviours.

The similarity of birds' songs to their fathers' songs was generally high across treatment groups, however noise did appear to specifically affect sequence similarity, or syntax, in both studies. Noise has been shown to affect certain aspects of song learning in previous studies due to auditory disruption (*Tschida & Mooney, 2012*; *Zevin, Seidenberg & Bottjer, 2004*). Traffic noise in particular has been shown to disrupt or mask other forms of parent–offspring communication in birds (*Leonard & Horn, 2008*; *McIntyre, Leonard & Horn, 2014*; *Schroeder et al., 2012*), therefore its impact on the accuracy of song learning, and especially the ability to copy long strings of syllables (even if the syllables themselves are accurate) is unsurprising. Zebra finch song is made up of common elements some of which

are also expressed as calls (*Price, 1979*). Hence, while the learning of individual elements is important for communication in general, the accurate sequencing of these elements is likely particularly important for song construction (*Menyhart et al., 2015*; *Riebel, 2009*; *Zann, 1993*). The fact that this characteristic was impacted by noise in both separate studies—and with two different "types" of low-frequency noise—therefore strongly indicates a significant disturbance to the song learning process in this species.

While we found that learning was impacted by noise, we found no evidence of the putatively adaptive changes in song that have been reported in wild populations living along urban-rural gradients (i.e., singing higher minimum frequencies in environments subject to anthropogenic noise). While at PHD 40, Study 1 birds in noise sang higher minimum frequencies, by day 100 the only effect of urban noise was on maximum frequency, which was slightly lower than in quiet treatment birds. In contrast, birds from Study 2 showed higher maximum and peak frequencies after chronic pink (1–3 kHz) noise exposure. Combined, these results are inconsistent and do not support the hypothesis that zebra finches alter their song in the long-term to improve transmission in a noisy environment. All birds were recorded in relative silence, which could mean that young birds were adjusting their song frequency to the current acoustic environment only (i.e., they may have sung at higher frequencies in the experimental chambers but not in the recording chamber). We do not know whether zebra finches possess the vocal flexibility to spontaneously alter the frequency of their songs, but it seems likely given that it has been observed in other species (*Potvin & Mulder, 2013*; *Verzijden et al., 2010*). We cannot rule out that there may have been other adaptive changes in the songs that reduced masking but that we did not detect. Nevertheless, we interpret our results to show that in this species, the masking of lower acoustic notes in the transfer from tutor to tutee, resulting in only higher notes being learned, is not the underlying mechanism by which acoustic adaptation occurs in this environment.

One unsuspected novel result from our study was our finding that the number of brothers an individual has may have an impact on song-learning accuracy. The number of siblings has been shown previously to affect some aspects of nestling condition (*Gil et al., 2006*) and mate preferences (*Holveck & Riebel, 2010*), however brood size did not appear to influence metrics of song learning in a previous study (*Gil et al., 2006*). A possible explanation for the effect of brood size on song learning that we observed is that more brothers may increase the noise in a nest, and therefore provide additional noise effects separate from already present chronic urban or background noise. This more immediate source of auditory disruption may limit the amount a juvenile bird is able to practice its song, leading to higher numbers of discrepancies among birds that have to compete with siblings. Zebra finches also require a sensorimotor phase whereby there is one-on-one interaction between tutor and tutee (*Derégnaucourt, 2011*); a large number of brothers could modify the nature of interactions between a bird and its father, and brothers may serve as potential tutors for each other, thus affecting the song learning process. Having many siblings has also been shown to negatively affect offspring quality (growth rate, biometry; *Gil et al., 2006*; *Potvin & MacDougall-Shackleton, 2015b*), which may in turn result in poorer song learning ability. Further investigation into how brood size or, more

specifically, the number of tutees in a group might impact song development over more specific time periods, especially through processes such as horizontal transfer between siblings, may shed more light on this finding.

We provide the first experimental findings for the impact of anthropogenic noise on song learning structures in the avian brain. We also found that noise affects the learning of song element sequences in particular. Both findings indicate that noise, along with brood size, is a crucial aspect of an individual's early environment with long-term consequences, despite noise not being identified as a physiological stressor. These results may also contribute to our current understanding of some of the difference in urban and rural birdsong. Of course, such conclusions do not rule out other processes that may be contributing to song changes in urban environments, such as sexual selection for effective urban songs or elements (*Candolin & Heuschele, 2008*). We suggest that future research focus on female preference of putatively urban-adapted song in urban and rural environments to disentangle whether sexual selection, rather than environmental pressures on song learning, might be the defining selective process behind song changes commonly observed in wild urban populations.

## ACKNOWLEDGEMENTS

We thank A Diez, M Rebuli, T Farrell, M Hasstedt, H An, A Boyer, and R Ellick for help with animal care and data collection.

### Funding

Funding for this project was provided by NSERC Canada grants to SAM-S, a TD Friends of the Environment grant to DAP, and a National Science Foundation grant IOS-1257590 to JPS. The funders had no role in study design, data collection and analysis, decision to publish, or preparation of the manuscript.

### Grant Disclosures

The following grant information was disclosed by the authors:
NSERC Canada.
TD Friends of the Environment.
National Science Foundation: IOS-1257590.

### Competing Interests

The authors declare there are no competing interests.

### Author Contributions

- Dominique A. Potvin conceived and designed the experiments, performed the experiments, analyzed the data, wrote the paper, prepared figures and/or tables, reviewed drafts of the paper.
- Michael T. Curcio performed the experiments, analyzed the data, wrote the paper, reviewed drafts of the paper.

- John P. Swaddle and Scott A. MacDougall-Shackleton conceived and designed the experiments, performed the experiments, contributed reagents/materials/analysis tools, wrote the paper, reviewed drafts of the paper.

### Animal Ethics

The following information was supplied relating to ethical approvals (i.e., approving body and any reference numbers):

All birds in Group 1 were kept and treated in accordance with guidelines set by the Canadian Council on Animal Care, and all procedures in this study were approved by the University of Western Ontario Animal Use Subcommittee (protocol number 2007-089). Group 2 protocols were approved by the College of William and Mary Institutional Animal Care and Use Committee (IACUC-2012-11-23-8173-jpswad).

### Data Availability

The raw data has been supplied as Supplemental Files.

### Supplemental Information

Supplemental information for this article can be found online at http://dx.doi.org/10.7717/peerj.2287#supplemental-information.

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
