# Peer review of "Experimental exposure to urban and pink noise affects brain development and song learning in zebra finches (Taenopygia guttata)"

_PeerJ, doi:10.7717/peerj.2287_

## Round 0.1 · original submission · Major Revisions

· Academic Editor

Major Revisions

I have now received three very helpful reviews from experts in your field. Although each reviewer notes several strengths of your study and each clearly respects the amount of work that went into this project, each also indicates some concern over the methods and resulting inferences. Reviewer 2 in particular raises some concerns about the validity of the "traffic noise" stimuli itself, which need to be addressed. Based on the clear strengths of the MS, I am willing to invite a revision, but you will need to expand on the detail of the methods and analyses according to the reviewers' questions. A revision will likely be sent out to the original reviewers. Thank you for submitting such interesting work to PeerJ.

Reviewer 1 ·

Basic reporting

This manuscript links two very interesting sets of ideas. One from urban ecology (acoustic adaptation) and one from neurethology (developmental stress affecting song learning). The work presented in this paper is truly integrative and bridges multiple fields of study. The idea being tested here is that urban noise results in developmental stress in fledgling Zebra finches, which results in impaired song learning. The authors incorporated many variables including multiple song characteristics, multiple recordings on different PHD, brain volume measurements, cortisol measurements and comparison between locations of testing and noise levels (soft, medium, loud). There are so many variables in this experiment and many controls described within the methods, including cross fostering. Because of this, the paper definitely needs a substantial improvement in clarity, especially statistical approaches. I needed to read it multiple times to understand the methods and it is still unclear whether the conclusions are truly supported by the data. I would be willing to reconsider a much revised version to re-evaluate the conclusions.

Introduction:
• There are many places throughout the introduction with very awkward wording that need improvement. Example: “We used male zebra finches (Taeniopygia guttata) that were bred
exposed, or not exposed, to traffic noise, recorded their songs and compared these to fathers’ songs”
Sentences like this require multiple readings and there are many in this paper. All sentences like this one should be located and improved.

• While there is mention of developmental stress affecting brain volume measurements, the authors do not discuss why they chose these regions. HVC, RA, AREA X are in the song system… So? Describe…
• On line 85 the description of alternative hypotheses linking noise to neuroanatomical changes: would be helpful to include song rate, feeding rate and # brothers in nest (discovered in this study) as possible explanations. Also, data on song rate and feeding rate should be added so the reader can evaluate whether these are related to brain changes.

Experimental design

Methods:
The methods are difficult to read. There is a lot to get through in this section. There are numerous places in which the reader gets side-tracked by the authors without proper explanation. For instance, the description of cross fostering. Describe in much more detail why this had to happen so it doesn’t come out of nowhere.
The biggest problem with the methods though is the lack of tables, timelines and illustrations. For instance, the PHD recordings can be placed on a timeline. The list of song characteristics examined can be listed in a table, even the cross fostering scheme can be in a table. A timeline could also include days of bleeding for CORT samples, days of recording adult songs, etc.
Number of songs the birds sang during the experiment should be estimated during an observation period. This will help reveal associations between song quality and quantity and HVC/Area X volume.
The brain tissue was sliced at 30um into 2 series and mounted onto slides. The volume measurements need much better description and how lost sections were accounted for. If there is a differences in sections that were lost, this could most certainly account for the treatment differences being reported. Much of this (as well as all the rationale for what brain regions were measured) was glossed over.
Statistical tests are hard to follow and I wonder why all these variables aren’t placed into a single model such as a DFA or a PCA. This was not a simple design and there are multiple variables here: multiple PHD recordings, multiple song characteristics, multiple locations of the experiment, CORT measures, brain measures, etc.
Can the authors put these into a single analysis to pull out the most important components. It is possible they did that with the regression but it is not easy to tell from the methods description or from the graph of the results.

Validity of the findings

The author mentions that # of brothers had an effect on brain volume of HVC/Area X. But there is no mention of whether this was different between treatment groups. Is # of brothers balanced between the treatments? This needs to be ruled out.
It needs to be clear that noise itself may likely not be the cause of brain volume differences. There are too many other factors that may be underlying this result. Possibly all other factors are influenced by noise (signing rate, quality of song, feeding rate that effect nutritional stress, # brothers, etc) but this paper does not elucidate any of these. Possibly a well-controlled study that only looked at whether neuroanatomical changes occur with noise and why might be needed.

Reviewer 2 ·

Basic reporting

No Comments

Experimental design

Doesn't match required standards (see author comments)

Validity of the findings

Problems with external validity (see author comments).

Additional comments

This study addresses the question whether chronic noise exposure induces stress in juvenile zebra finches and, in turn, affects their brain development and song learning. The manuscript is generally very well written and I enjoyed reading it. However, I also had several concerns.

Given that the manuscript deals with the effects of traffic noise I was hoping to read more about the nature of the noise playback. For instance, how exactly was the traffic noise recorded? What were the noise sources in the habitat and how far from them was the recording microphone positioned? This information is crucial because the entire experiment hinges on the noise playback. Moreover, the manuscript forgot to mention the duration of the used noise file. Traffic noise typically fluctuates and if you happen to pick a moment of unusually high or low intensity, the playback would not mimic typical traffic noise. Moreover, and most importantly, it is unclear why the traffic noise recording was mixed with various other noises downloaded from the internet. The problem with this is that it is unclear how these sounds were recorded and thus it is impossible to tell how relevant they are in terms of the hypothesis tested. Furthermore, the manuscript does not mention how the sounds were edited and with which relative amplitudes they were mixed, which makes it even more difficult to assess the external validity of the experiment.
By the end of the day, the study does not seem to test the effects of traffic noise but the effects of an artificial noise that was synthesized by the experimenter. The same is true for the noise used in Group 2: constant pink noise is not traffic noise, neither in terms of the spectral shape of the noise nor the temporal fluctuation patterns. Moreover, in contrast to the constant noise exposure used in the experiment, traffic noise levels are typically fluctuating in the short-term and in addition they are usually lower during the night than during the day.
Given the experimental design of the study, a valid conclusion would be that chronic exposure to loud sounds can affect song learning and brain development in zebra finches but inferences about traffic noise do not seem warranted.

I take it that juveniles from different pairs could hear each other in the recording chamber/room. Thus, not only fathers are potential tutees but also the males in the other cages. Moreover, young zebra finches may also learn songs from other juveniles (Derégnaucourt & Gahr 2013 Biol Lett 9: 20130247). Thus, the clutches in each chamber are not statistically independent replicates. Considering this caveat, the sample size of Group 1 is N = 3 noise chambers and that of Group 2 is N = 1 noise room.

It would be good to know how loud the noise was at the position of the birds (i.e. the position of the perches). Given the huge variation in SPL levels between the cages I assume that the loudspeakers were positioned relatively closely to the cages. If this is true, then the actual SPL levels at the position of the birds was most likely much higher than the values reported for the center of the cage.

The fathers’ songs at an offspring age of 200 days post hatching in Group 2 does do not seem appropriate for the tutor-tutee comparisons because the offspring will not learn these songs. Zebra finches have their sensory phase of vocal production learning about 15-60 days posthatch, and therefore the tutor songs produced during this period should be used as reference for the song learning trajectory.

Given that the ambient noise in the recording room was 50 dB and the measuring threshold for minimum and maximum frequencies was -30 dB below peak amplitude, the songs need to have an amplitude of at least 80 dB SPL to yield reliable measurements, right? To assess this issue, one needs to know the signal-to-noise ratio of the recordings. Especially subsong is low in amplitude and depending on the position of the microphone, the recorded amplitude might have been lower than 80 dB, which might have compromised the frequency measurements.

Another concern is related to the corticosterone measurements. Why were the birds sampled as adults when song learning was completed? Given the hypothesis of the study that noise induced stress impairs learning, one needs to measure corticosterone levels during the sensorimotor period, i.e. before day 90 post hatch.


Minor points:

Line 85: I understand that impaired auditory feedback by low-frequency noise may lead to higher frequency songs (line 103). But why should it increase song variation?

Line 132: How were the noise levels measured exactly? Please provide sufficient information to enable others to replication the work.

Line 133: In addition to average values, the range of noise levels would be interesting, too.

Line 156: I suppose the noise was switched off to record the offspring songs, right? Otherwise the noise would mask the songs (especially subsong) and impair the frequency measurements.

Line 252: “visual discrimination of the spectral space in the recording occupied by the song, using power spectra and confirmation with spectrograms” is difficult to understand. Please mention how exactly the measurements were made.

Line 263: which hemisphere was sectioned? By measuring the nuclei from one hemisphere only the study assumes that there was no size difference in HVC, Area X, and RA between hemispheres. To convince readers that measuring only one side is appropriate it would be good to show that the nuclei were of similar size in both hemispheres.

Line 275: similar published studies often report the brain mass and I think this would also be interesting in this study.

Line 444: or it may be the result of horizontal song transmission between brothers (see first major comment).

Reviewer 3 ·

Basic reporting

please see comments to authors below

Experimental design

please see comments to authors below

Validity of the findings

please see comments to authors below

Additional comments

The authors conducted two separate experiments to determine the effects of played-back traffic noise on song learning and brain region development in young zebra finches. Both studies found effects of noise on these processes/structures, though not always in predictable ways. No effect of noise playback on circulating levels of corticosterone were detected. Syllables of songs did not appear to be affected by noise exposure, but young birds exposed to noise were less likely to copy well the ordering of syllables of their particular tutor song models. Area X and HVC regions were smaller in birds exposed to noise.

This is an exciting and strong study. The authors have done an impressive amount of work in carrying out two separate experiments (it would be nice to see more authors do this sort of within-study ‘replication’ of an experiment). Also, the authors have assessed a wide range of dependent variables in these experiments, all converging on the important questions about song learning and production. To top it all off, the manuscript is written really well and clearly.

I do have a few questions / concerns about the work, which I raise here by line number and not in any order of importance.

42-48 Here and perhaps later in the discussion, the authors might wish to consult the experimental study of Owens et al. (2012, Behavioural Processes). Those authors found that extended exposure to traffic noise playbacks resulted in members of chickadee and titmouse flocks associating more closely with one another in space. Presumably a key way to solve the problem of calls or songs being masked by noise in these social species is to ‘clump’ closer together in space. On this note, do the authors have any information about actual singing behavior and interactions of the zebra finch males in their study, beyond the recorded songs themselves? Perhaps ‘noise’ males were singing or interacting socially with others differently than the control males?

60 I could not find Wright et al. 2007 in the reference section

66 How are the authors defining ‘complexity’ here? If they mean diversity of sounds produced, that might be the better way to make the point here – ‘complexity’ can mean a wide range of things to different readers!

127-131 The authors should provide more detail on the nature of this traffic noise playback. Was there just one variant of traffic noise that was played back to everyone? If so, how do the authors get around the ‘simple pseudoreplication’ argument raised by Kroodsma, McGregor, and others over the years? How long was the variant / were the variants? What was the nature of the traffic noise being ‘combined with soundtracks of trains, cars, motorcycles, and lawnmowers...’? That is, were these added at certain rates? How many of each different kind were added? Since the traffic noise manipulation is the key experimental manipulation in the study, more description of exactly what was played back would be helpful.

201-202 Study 2 noise playback – many of the same issues as raised above! In addition, although pink noise or manipulated white noise are frequently used as synthesized versions of traffic noise, are there any validations of this manipulation? To my knowledge, no one has assessed whether these simulated noise playbacks are perceived by the study subjects as traffic noise, or as simply a novel stimulus in their environment. Perhaps these vocal responses to ‘traffic noise’ that have been reported in the literature are more general neo-phobic vocal responses to novel stimuli in the environment?

224-256 Can the authors report inter-observer reliability assessments for the behavioral coding here? I realize a lot of their analysis of song was automated in Raven, but it looks like some of the work (e.g., lines 230-233, 249-250) involved researcher coding, and so there should be some data provided on inter-rater agreement.

275 On the note raised above, it was really nice to see that the authors coded these brain regions blind with regard to experimental group!

288-291 It seems safe to assume that many of the acoustic parameters of songs and syllables of these birds are correlated with one another. Perhaps for the many acoustic parameters, the authors could perform a PCA or factor analysis, and then run the statistics on those PCs or factors that emerge after reducing the data?

330-334 The authors might double-check the results presented in Figure 1 here – I am not sure the data presented in the figure back up these two sentences. For instance, it does not look like there is a 20% drop in maximum frequency for the ‘High noise’ group compared to the two lowest noise groups. Also, error bars for peak frequency in the ‘High noise’ group appear to overlap broadly the error bars of the lower noise groups.

338-339 The authors should really check the results plotted for Study 2 birds in Figure 1 – it looks to me like the means and error bars for ‘Silent’ and ‘Noise’ are virtually identical, if not exactly the same.

392-399 and Figure 2 Why was the HVC effect the strongest in the ‘Moderate’ noise group as opposed to the ‘High’ noise group? In fact, this seems like the largest effect detected in the study, based upon Figure 2! This makes no biological sense.

402-403 Do the authors have this information on ‘reduced singing behaviour’?

433-438 My understanding of crystallized / stereotyped song in zebra finches is that birds would NOT be able to adjust the acoustics of their songs like this. Is there more flexibility in adult song? Can the authors cite studies in zebra finches to back up this suggestion?

Figure 3 Another odd finding is that, while there is a nice linear-like decrease in sequential similarity in song as noise levels increase, birds in the ‘Moderate’ noise group are again providing outlier data. Was there something odd about the cage setting for birds in this group? Is there potentially a confound here that might explain the odd data for birds in this group?

---

## Round 0.2 · Minor Revisions

· Academic Editor

Minor Revisions

With the previous round of reviews, one reviewer was quite positive about your MS whereas two reviewers had significant concerns. Unfortunately, I was not able to obtain reviews again from both of the more critical reviewers and I have arrived at a difficult place where the remaining two reviewers have quite disparate opinions about your MS. Typically, I would defer to the more critical reviewer, especially given that both reviewers are more expert in this particular area than am I. However, I would like to give you the opportunity to address the remaining criticisms because I believe that there is much that is worthwhile in your rather ambitious study and my feeling is that data generally deserves to be published as long as one is suitably circumspect in the conclusions that one draws from it, and that it might serve to encourage others to continue more detailed study of the topic. Therefore, I invite you to revise the paper again being sure to include your responses to the reviewers’ concerns within the MS itself and not just within your response letter.

I agree with Reviewer 2 that it would be nice if you could go further in terms of explaining the observed effects and lack of effects. However, to my mind, the absence of data on feeding and singing rates does not preclude the publication of the current findings. I think it is sufficient to point to the need for future studies to follow up on what you have presented here, which you have done. No single study should be expected to answer all possible questions.

Furthermore, I recognize the reviewer’s concern that the traffic noise samples were not solely comprised of traffic noise. I think this is ok in terms of ecological validity because, as you point out, in nature, one does not hear traffic sounds in isolation. However, it does make it questionable whether you are really speaking about traffic noise or noise in general; this point of the reviewer’s is legitimate. Given that your findings are similar for your “pink noise” condition, I wonder if this actually adds to the reviewer’s claim that you are not measuring traffic noise per se. You have done a good job accounting for this criticism in your revision but you might go even further to make more general comments about “noisy environments” where you instead still focus on traffic noise (e.g. line 136).

In addition, Reviewer 2 has other concerns that I would need to see addressed before I could recommend publication. For instance,
1. The manuscript still does not mention how the “traffic” noise files were edited and with which relative amplitudes the additional sounds were mixed. The reference to Potvin et al. (2015 Anim Behav 107:201-207) doesn’t help because this paper doesn’t provide the necessary information either.

2. the manuscript still does not give any information about the dimensions of the cages and the sound booths because possible noise variation inside the cages could be inferred from this, provided that the exact position of the loudspeakers is known. Again, the reference to Potvin et al. (2015) in the manuscript isn’t helpful because the cited paper does not mention the chamber and cage sizes.

3. it cannot be ruled out that the birds in this study copied songs across cages (the reference to Mann & Slater 1994 and Mann & Slater 1995 is a bit misleading because neither of the studies actually tested whether cross-cage learning occurred. Mann & Slater (1995) didn’t even use caged pairs.) Therefore, I still think the manuscript either needs to show that the birds in each booth didn’t copy syllables from other cages or treat each booth as one data point.
This last comment suggests that you need to use different references here. But, further, could you conduct a multi-level model regression analyzing the data at the level of both individual and booth to assuage these concerns?

It should be possible to provide the cage dimensions and lay-out in a revision.

The reviewers previously noted concerns about the use of multiple predictors. Although you dismissed the idea of data reduction – could you conduct commonality analyses to reveal any shared or common effects of the predictors? Kim Nimon has provided script in R and SPSS that is helpful.

Reviewer 3 also makes numerous suggestions for editing that I hope you will attend to in the revision.

I have some additional edits of my own:

On line 39, you are missing ‘been” at the end of the sentence.
Please edit for American spelling throughout e.g. behavior instead of behavior.
Please use APA style consistently for references (e.g. Use & instead of “and” within parentheses)
Is it really necessary that cognitive development is affected, resulting in altered songs (lines75-76), rather than an explanation that rests solely upon adapting to differing noise levels in the environment when learning? If feedback is negatively impacted by noise, I’m not sure this suggests a developmental or neurological response. Indeed, you suggest other mechanisms later in the introduction (e.g. lines 95-97) so I think the hypothesis that there are neurological or cognitive changes as a result of noise in the environment could be toned down. It seems that you yourselves argue against such changes in the discussion.
In the introduction it should be clear why you will be presenting two studies; one with traffic and other urban noise and a second with pink noise. It isn’t clear why you conducted both studies from the introduction. Nor does it become much clearer in the discussion.
Try to avoid describing your subjects as having “been used” for the experiments.
Please insert a reference on line 348.
I look forward to seeing a revision of your MS. I do not think that I will need to send it out for further review.

Reviewer 2 ·

Basic reporting

see General Comments

Experimental design

see General Comments

Validity of the findings

see General Comments

Additional comments

The revised manuscript has been considerably improved but there are still a number of unresolved issues. Moreover, the revision also exposed the limitations of the study.

It is unfortunate that the manuscript cannot provide data on singing and feeding rates, as both of these factors have been shown to affect song learning in zebra finches and both might be affected by noise exposure. Without this data, the study misses the opportunity to really find out what is behind the observed effects and the conclusions of the manuscript thus remain cursory.

The manuscript still does not mention how the “traffic” noise files were edited and with which relative amplitudes the additional sounds were mixed. The reference to Potvin et al. (2015 Anim Behav 107:201-207) doesn’t help because this paper doesn’t provide the necessary information either.
Anyway, the revision could not dispel the concern regarding the validity of the noise playback. The noise file contains traffic sounds but it is not a recording of natural traffic noise and thus conclusions about real world traffic are not warranted. This distinction is important because if the study would have tested real traffic noise, it would be relevant regarding conservation and urban planning but based on an unnatural “super noise”, with various other noises added on top of real traffic noise, the study has only a very limited ecological validity.
It is not clear why instead of using the traffic noise recordings from Melbourne, these recordings were mixed with various other sounds, including lawnmowers (!). Obviously, this is not a typical traffic noise. Moreover, using sound clips downloaded from a website is problematic when, as in this case, it is unknown how and with which equipment the recordings were made. At the end of the day, what the study used is not traffic noise but a sound file of high density noise synthesized by one of the authors.

It is true that zebra finches prefer live tutors - yet they also learn songs from inanimate sources, e.g. tape (Adret 1993 Anim Behav 46:149-159, Houx & ten Cate 1999 Anim Behav 57:837–845, Funabiki & Funabiki 2009 Dev Neurobiol 69:752–759). In addition, tutees often learn songs from more than one adult male (Williams 1990 Anim Behav 39:745-757, ten Cate & Slater 1991 Anim Behav 42:150-152). With a typical learning success from tape of about 60% (Deregnaucourt et al. 2013 J Physiol Paris 107:210–218), it cannot be ruled out that the birds in this study copied songs across cages (the reference to Mann & Slater 1994 and Mann & Slater 1995 is a bit misleading because neither of the studies actually tested whether cross-cage learning occurred. Mann & Slater (1995) didn’t even use caged pairs.) Therefore, I still think the manuscript either needs to show that the birds in each booth didn’t copy syllables from other cages or treat each booth as one data point.

The revised manuscript still lacks information about how much the noise varied inside the cages. I was hoping to see actual measurements and not just a brief mentioning that noise variation was negligible. In this context it is unfortunate that the manuscript still does not give any information about the dimensions of the cages and the sound booths because possible noise variation inside the cages could be inferred from this, provided that the exact position of the loudspeakers is known. Again, the reference to Potvin et al. (2015) in the manuscript isn’t helpful because the cited paper does not mention the chamber and cage sizes.
Telling from the picture on the AFAR webpage (http://birds.uwo.ca/facilities/index.html), the sound booths are rather narrow and I wonder how the cages were arranged in them. Where they stacked on top of each other? Neither this manuscript nor the previous paper using this set up (Potvin et al. 2015) makes any mention of the arrangement of the cages. If they were indeed stacked on top of each other, then this raises the concern that the noise treatments are potentially confounded by the cage position within the booth.

Reviewer 3 ·

Basic reporting

No Comments

Experimental design

No Comments

Validity of the findings

No Comments

Additional comments

The authors have done a fine job of revising their manuscript to address questions and concerns raised by reviewers. I only have a few fairly minor (and mostly stylistic) suggestions regarding the revision.

Line 37 ‘syntax...was affected by noise.’ – can the authors make this point about the effect of noise directional, as in whether noise led to more or less variation in ordering of syllables?

Line 39 replace ‘had’ with ‘were’ to read ‘...males that were exposed...’

Line 84 delete ‘in birds’ as ‘birds’ appears in the previous line.

Line 87 to retain parallel form in these phrases, replace either ‘interruption’ with ‘interrupting’ or ‘impairing’ with ‘the impairment of’.

Line 128 instead of ‘less complex’, specify what is meant – fewer syllable types in songs?

Line 167 replace ‘described above’ with ‘described below’.

Lines 402-403 suggest placing this sentence later in the Results section, so that the section does not start off with a non-significant finding. Perhaps it could go just before the paragraphs on neural regions that were assessed?

References Some have first letter caps for all words – Blickley et al. 2012, Dooling & Blumenrath 2013, Gavett & Wakeley 1986, etc.

---

## Round 0.3 · accepted · Accept

· Academic Editor

Accept

Thank you for being attentive to this last round of comments on your very interesting MS. I am now happy to accept your paper for publication and thank you for your patience with this process. I think the findings are now much clearer and the increased clarity will positively affect the impact of the work.